# Optimization Algorithms for Joint Power and Sub-Channel Allocation for NOMA-Based Maritime Communications

**DOI:** 10.3390/e23111454

**Published:** 2021-11-01

**Authors:** Huanyu Li, Hui Li, Youling Zhou

**Affiliations:** 1Department of Information and Communication Engineering, Hainan University, Haikou 570228, China; 19081000210009@hainanu.edu.cn (H.L.); zhouyl@hainanu.edu.cn (Y.Z.); 2Department of Electronic Information Engineering, Binjiang College of UNIST, Wuxi 214105, China

**Keywords:** offshore communications, non-orthogonal multiple access, power allocation, maritime user allocation, joint resource optimization

## Abstract

This paper investigates resource optimization schemes in a marine communication scenario based on non-orthogonal multiple access (NOMA). According to the offshore environment of the *South China Sea*, we first establish a Longley–Rice-based channel model. Then, the weighted achievable rate (WAR) is considered as the optimization objective to weigh the information rate and user fairness effectively. Our work introduces an improved joint power and user allocation scheme (RBPUA) based on a single resource block. Taking RBPUA as a basic module, we propose three joint multi-subchannel power and marine user allocation algorithms. The gradient descent algorithm (GRAD) is used as the reference standard for WAR optimization. The multi-choice knapsack algorithm combined with dynamic programming (MCKP-DP) obtains a WAR optimization result almost equal to that of GRAD. These two NOMA-based solutions are able to improve WAR performance by 7.47% compared with OMA. Due to the high computational complexity of the MCKP-DP, we further propose a DP-based fully polynomial-time approximation algorithm (DP-FPTA). The simulation results show that DP-FPTA can reduce the complexity by 84.3% while achieving an approximate optimized performance of 99.55%. This advantage of realizing the trade-off between performance optimization and complexity meets the requirements of practical low-latency systems.

## 1. Introduction

The rapid development of the blue economy has created the challenge of constructing intelligent ports and terminals. At the same time, this has led to a significant increase in demand for maritime digital services from offshore users. Compared with traditional satellite communications, shore-based base stations (BS) can provide low-cost, high-speed mobile communications services. Recently, several references have applied fifth-generation (5G) and even sixth-generation (6G) emerging technologies, such as massive multiple-input multiple-output (MIMO) [1], internet of vehicles [2], and mobile edge computing [3] to maritime communications. Considering the limited geographic availability of coastal BS, this paper investigates resource allocation schemes based on non-orthogonal multiple access (NOMA) in 5G. Our contribution is to provide high data rate services for more maritime users (ships and islands) in offshore areas with limited-spectrum resources.

According to previous research, the maritime communication environment is characterized by instability, sparseness, and the evaporation ducting effect [1]. These features make the marine channel model different from the terrestrial channel model. In particular, due to the scarcity of obstructions at sea, the shadow effect hardly affects signal transmission loss. Furthermore, we can neglect the evaporation ducting phenomenon, since offshore communication takes place within the line of sight (LoS) [4]. The fading of the offshore channel can be divided into large-scale fading and small-scale fading [5]. First, it is essential to establish an appropriate large-scale fading model. The radio communications sector of the international telecommunication union (ITU-R) model fails to consider sea surface reflection and other complex marine environmental factors [6]. Previous studies [7] have modified the classic Okumura–Hata model by considering the sparse distribution of scattering in the context of vast ocean areas. The Longley–Rice model (also known as ITM) includes marine environment and climate factors and the deployment of antennas [8,9]. Hence, ITM is able to achieve a more accurate simulation conclusion than the Okumura–Hata model [8]. In addition, it is also necessary for the maritime channel model to consider small-scale fading caused by sea surface fluctuations and atmospheric scattering [10,11]. The research of [12] shows that the probability distribution function of small-scale fading is closer to a Rice distribution than a Nakagami distribution or Rayleigh distribution. In this paper, we establish a Longley–Rice-based offshore channel model that includes Rice small-scale fading.

Joint power and subchannel allocation have always been the focus of resource optimization in NOMA. Some works also refer to subchannel allocation as the allocation scheme for multiplexed users on each resource block (RB). Motivated by recent research, our work takes the weighted achievable rate (WAR) as an objective function to weigh the WAR performance and user fairness [13]. We summarize NOMA-based resource optimization schemes into the following two cases according to the power constraints (user power constraint and system power constraint).

Case 1: User power constraint is generally applied to the power control for each user in uplink communications [14,15]. However, some papers still set user power constraints for downlink communication scenarios [16,17]. The results of [17] prove that the WAR optimization problem with user constraints is strongly non-deterministic polynomial hard (NP-hard). Therefore, researchers developed a Lagrangian dual algorithm to achieve an approximate optimal result. However, the computational difficulty cannot be ignored in a low-latency actual system.

Case 2: The total system power constraint represents the power budget provided by the base station to the communication system. The authors of [18] prove that the equal-weighted achievable rate optimization is solvable by polynomial time. Nevertheless, this does not mean that the WAR optimization problem is also solvable. Several papers have addressed this complex WAR maximization problem. The authors of [17] propose a two-stage dynamic programming scheme that can achieve an optimal WAR at the cost of higher computational complexity. Aiming at a series of non-convex combination optimization problems, the authors of [19] develop a joint user and power allocation scheme based on convex programming differences. Moreover, the authors of [20] propose a bilateral matching scheme derived from game theory to solve the problem of joint subchannel and power allocation. This method is extended to optimize the user and sub-channel pairing problems occurring in [21,22]. The work of [23] exploits dual theory and the sub-gradient method to maximize the data rate under the NOMA-D2D background. In NOMA-based cognitive radio (CR) networks [24,25], researchers have put forward some effective resource optimization plans to maximize the achievable throughput in the network. The authors of [26] developed a monotonic optimization scheme to maximize the weighted system throughput. Due to the high complexity of this program, it can only be regarded as an optimization reference index. However, the solutions provided in [27] reduce the complexity by discretizing the power value to achieve the approximate optimal performance.

According to the summarization of our reference survey, the research reviewed barely considers the polynomial-time approximation (PTA) algorithm. Nevertheless, this kind of approximate optimization scheme can adjust the calculative complexity according to the optimization requirements. Inspired by the above observations, we propose the use of a fully polynomial-time approximation (FPTA) algorithm for WAR optimization in the downlink scenario of a NOMA-based offshore communication system. The contributions of this work are as follows:This work applies the NOMA-based resource optimization schemes to offshore communication in the *South China Sea*, improving the spectrum efficiency and data rate of the system. We propose an appropriate channel model based on the Longley–Rice model, including environmental factors and antenna deployment. Our offshore channel can be widely used in various regions of the world just by modifying its parameters.We first analyze the resource optimization strategy using a single resource block. Aiming at the problem of single-resource-block joint power and user allocation, we implement an improved scheme (RBPUA) with higher computational efficiency.Taking the single-resource-block optimization as the basic module, we propose an efficient joint multi-resource-block optimization scheme. This scheme combines low-complexity RBPUA with the gradient descent algorithm (GRAD). The simulation results show that GRAD with an accuracy of 0.0001 can achieve an optimal WAR performance. Moreover, the WAR of the NOMA maritime system is significantly improved compared with orthogonal multiple access (OMA).An innovative solution is introduced for multi-resource-block NOMA maritime communication, which is derived from the multiple-choice knapsack algorithm and dynamic programming (MCKP-DP) [28]. The simulation consequence proves that the MCKP-DP scheme is able to obtain an optimal WAR nearly equal to that of the GRAD algorithm.Inspired by the MCKP, we further develop an FPTA optimization algorithm based on dynamic programming (DP-FPTA). The scheme can perfectly balance the WAR performance optimization and computational complexity by adjusting the error parameter ε. Therefore, DP-FPTA is more suitable for practical NOMA maritime communications with strict latency requirements.

The rest of this paper is organized as follows. In Section 2, we first introduce the improved maritime channel model and system parameters, followed by the construction of a NOMA-based downlink offshore communication system. Section 3 formulates the resource optimization problem with WAR as the objective function. In Section 4, we propose an improved joint optimization scheme on a resource block RBPUA with high computational efficiency. Taking RBPUA as the basic module, Section 5 develops three joint multi-subchannel power and marine user allocation algorithms. Section 6 presents the simulation results. The WAR optimization performance and the calculating complexity of our proposed schemes are highlighted for comparison. Lastly, we discuss and conclude our work in Section 7.

## 2. System Model and Parameters Analysis

This section discusses the maritime channel model and network setting for the downlink NOMA offshore communication system. At the end of this chapter, we consider the objective function of the resource optimization problem.

With reference to the offshore communication environment of the *South China Sea*, this paper constructs a Longley–Rice-based channel model that obeys a Rice distribution. The reference value for the Longley–Rice transmission loss is expressed as follows:(1)$$L_{cr}\left(dB\right)=L_{bf}+A_{cr}\;\mathrm{dB},$$
where the formula for fundamental transmission loss *L_bf_* in free space is:(2)$$L_{bf}\left(\mathrm{dB}\right)=32.45+20\log_{10}f+20\log_{10}d\;\mathrm{dB},$$
where the distance *d* is in km and the radio frequency *f* is in MHz. It is worth noting that the Longley–Rice model is suitable for signal propagation with carrier frequencies between 20 MHZ and 20 GHz.

The reference attenuation *A_cr_* relative to free space depends on the communication distance:(3)$$A_{cr}=\{\begin{array}{c} max(0,A_{e}+k_{1}d+k_{2}\log_{10}d),(dLoS_{min})\\ A_{d}+m_{d}d,\left(d_{LoS}\leq d\leq d_{x}\right)\\ A_{s}+m_{s}d,\left(d_{x}\leq d\right) \end{array}\;\mathrm{dB}.$$

We conclude from Equation (3) that the Longley–Rice transmission loss model is divided into three cases according to the size of the communication range: the LoS model $(dLoS_{min}$), the diffraction model $\left(d_{LoS}\leq d\leq d_{x}\right)$, and the scattering model $\left(d_{x}\leq d\right)$ [8]. In Equation (3), *d_x_* indicates the boundary point where the diffraction loss and the scattering loss are equal. *A_e_*, *A_d_*, and *A_s_* (unit: dB) represent the propagation loss of LoS, diffraction, and scattering in free space, respectively. Moreover, *k_1_* and *k_2_* represent the propagation loss coefficients. *m_d_* and *m_s_* are the diffraction and scattering loss coefficients, respectively. Due to the limited coverage of a 5G BS, our communication range between coastal BS and ships is within the horizon. Therefore, our offshore channel needs to only consider the LoS model. It is worth noting that the applicable transmission range of the Longley–Rice model specifies a lower bound *d_min_* (generally set to 1 km). Enlightened by the guidebook [29], our channel model can be regarded as the fundamental transmission loss in free space (Equation (2)) within 1 km.

The calculative process of the reference attenuation *A_cr_* is very complicated. In that case, we have to consider the following essential parameters: climate factor (1–7), terrain irregularity parameter Δ*h* (Table 1), the effective height of the transmitter/receiver antennas *H_T_*/*H_R_*, the system deployment parameters (*very careful*, *careful*, *random*), the conductivity and permittivity of the seawater (Table 2), and the surface refractivity (Table 3). The mature Longley–Rice model provides seven climate backgrounds: (1) *equatorial*; (2) *continental subtropical*; (3) *maritime tropical*; (4) *desert*; (5) *continental temperate*; (6) *maritime temperate* (*over land*); and (7) *maritime temperate* (*over sea*). These seven kinds of climates cover the communication background of the global offshore ports, meaning that our model is the most widely used. The irregular terrain parameter Δ*h* indicates the degree of sea wave undulation. Table 1 presents the parameter values given by the Longley–Rice model for different types of irregular terrain. In addition, the system deployment parameter reflects the qualitative description of the site selection scenario for each terminal, which will impact the signal strength around the terminal and the effective height of the antenna. According to the simulated geographic environment of the *South China Sea*, we will specifically discuss the assignment of the above parameters in Section 6. For the detailed calculation process of their participation in *A_cr_*, please refer to [8] and [29].

As shown in Figure 1**,** a downlink NOMA marine system is composed of a shore-based base station in the offshore area of the *South China Sea*. We use T≜{1,2,…,T} to describe the index set of maritime users. This maritime system divides the entire bandwidth *B* into *S* subchannels, where each bandwidth is *B_S_*. This paper refers to a subchannel as a resource block. We adopt S≜{1,2,…,S} to represent the set of these resource blocks; for each resource block, s∈S satisfies $\sum_{s\in S}B_{S}=B$. In our maritime system, the resource blocks are of the orthogonal frequency division to ensure that they will not interfere with each other.

Suppose that $P_{t}^{s}$ represents the transmitting power from the coastal BS to a maritime user t∈T on resource block *s*. If $P_{t}^{s}>0$, this means that the maritime user *t* obtains power from resource block *s*. Therefore, we determine that user *t* is valid on this resource block. In contrast, $P_{t}^{s}=0$ indicates an offshore user that does not occupy resource block *s*; we define this as an invalid user. On resource block *s*, let $G_{t}^{s}$ denote the channel gain from BS to maritime user *t* and $\sigma_{t}^{s}$ represent the noise power. To facilitate subsequent simplification, we define a normalized channel noise parameter: ${\tilde{\sigma }}_{t}^{s}≜\frac{\sigma_{t}^{s}}{G_{t}^{s}}$. In this manner, we reduce the number of variables affecting the optimization objective function. Let $P≜{\left(P_{t}^{s}\right)}_{s\in S,t\in T}$ indicate the matrix of the transmitting powers provided by BS for each maritime user. Moreover, $P_{s}≜{\left(P_{t}^{s}\right)}_{t\in T}$ denotes the vector of powers allocated to each maritime user on resource block *s*.

Successive interference cancellation (SIC) technology is applied in NOMA systems to effectively eliminate signal interference among users. With the increase in the number of multiplexed users, the SIC decoding process becomes very complicated [7]. Therefore, the upper limit of users carried on each resource block is defined as *A*. We use $A_{s}≜\{t\in T,P_{t}^{s}>0\}$ to denote the set of valid users on resource block *s*. Then, the above restraint condition can be summarized as |*A_s_*| ≤ *A*, where |·| represents the cardinality of a limited set.

Referring to the SIC decoding principle described in [30,31], we denote the SIC decoding order of *T* maritime users as *β_s_*: {1, 2,..., *T*}. The index of the *m*th decoded maritime user is *β_s_*(*m*) on resource block *s*, and the inverse function *β_s_^−^*^1^(*t*) indicates the decoding sequence of user *t*. The optimal decoding sequence is determined as Equation (4) according to ${\tilde{\sigma }}_{t}^{s}$ from highest to lowest:(4)$${\tilde{\sigma }}_{\beta_{s}\left(1\right)}^{s}\geq {\tilde{\sigma }}_{\beta_{s}\left(2\right)}^{s}\geq \cdots \geq {\tilde{\sigma }}_{\beta_{s}\left(T-1\right)}^{s}\geq {\tilde{\sigma }}_{\beta_{s}\left(T\right)}^{s}.$$

The Shannon Equation (5) enables us to solve the achievable rate of maritime user *t* on resource block *s*:(5)$$C_{t}^{s}\left(P_{s}\right)=B_{S}\log_{2}(1+\gamma )\;\mathrm{bit}/s,$$
(6)$$\gamma =\frac{G_{t}^{s}P_{t}^{s}}{\sum_{m=\beta_{s}{}^{-1}\left(t\right)+1}^{T}G_{t}^{s}P_{\beta_{s}\left(m\right)}^{s}+\sigma_{t}^{s}}=\frac{P_{t}^{s}}{\sum_{m=\beta_{s}{}^{-1}\left(t\right)+1}^{T}P_{\beta_{s}\left(m\right)}^{s}+\frac{\sigma_{t}^{s}}{G_{t}^{s}}}=\frac{P_{t}^{s}}{\sum_{m=\beta_{s}{}^{-1}\left(t\right)+1}^{T}P_{\beta_{s}\left(m\right)}^{s}+{\tilde{\sigma }}_{t}^{s}}.$$

## 3. Problem Formulation

We use **α** = {*α*_1_, *α*_2_,..., *α_t_*} to represent the sequence set of *T* positive weights. Furthermore, we introduce a weighting factor for each maritime user *t*. Then, we analyze the property of the objective function according to the weight value. By introducing the weight value, we can solve the performance optimization and user fairness based on multiple resource blocks [13]. The issue *W* aims at maximizing the WAR of the NOMA maritime system:(7)$$W=max\sum_{t\in T}\alpha_{t}\sum_{s\in S}C_{t}^{s}\left(P_{s}\right).$$
(8)$$w1:\sum_{t\in T}P_{t}^{s}\leq P_{max}^{s};$$
(9)$$w2:\sum_{t\in T}\sum_{s\in S}P_{t}^{s}\leq P_{max};$$
(10)$$w3:\left|A_{s}\right|\leq A;$$
(11)$$w4:P_{t}^{s}\geq 0.$$

It is noteworthy that Condition *w*1 restricts the upper power limit of each resource block. In Condition *w*2, *P*_max_ represents the upper bound of the total power provided by coastal BS. Condition *w*3 reflects the finite multiplexed maritime users carried by each resource block: |*A_s_*| ≤ *A*. Additionally, Constraint *w*4 ensures that the offshore user obtains a non-negative allocated power.

Problem *W* can be solved by polynomial decomposition to be equivalent to *W**:(12)$$W^{*}=max\sum_{s\in S}\sum_{n=1}^{T}\varphi_{n}^{s}\left({\tilde{p}}_{n}^{s}\right)+C.$$
(13)$$w1^{*}:{\tilde{p}}_{1}^{s}\leq P_{max}^{s};$$
(14)$$w2^{*}:\sum_{s\in S}{\tilde{p}}_{1}^{s}\leq P_{max};$$
(15)$$w3^{*}:\left|A_{s}*\right|\leq A;$$
(16)$$w4^{*}:{\tilde{p}}_{n}^{s}\geq {\tilde{p}}_{n+1}^{s},n\in \left\{1,\ldots ,T\right\};$$
(17)$$w5^{*}:{\tilde{p}}_{T+1}^{s}=0.$$
where *C* refers to a constant term formula separated from *W*:$C=\sum_{s\in S}\alpha_{\beta_{s}\left(T\right)}{log}_{2}(\frac{1}{{\tilde{\sigma }}_{\beta_{s}\left(T\right)}^{s}})$. In the following discussion, we leave the constant *C* aside and investigate the influence of function $\varphi_{n}^{s}$ on the optimization results. The modification problem *W** displays a detachable target function in dimensions n∈{1,2,…,T} and s∈{1,2,…,S}, respectively. In addition, function $\varphi_{n}^{s}\left({\tilde{p}}_{n}^{s}\right)$ depends on the variable of ${\tilde{p}}_{n}^{s}$ only to simplify the optimization workload.
(18)$${\tilde{p}}_{n}^{s}=\{\begin{array}{c} \overset{T}{\underset{m=n}{\sum }}P_{\beta_{s}\left(m\right)}^{s},n\in \left\{1,\ldots ,T\right\}\\ 0,n\geq T+1 \end{array},$$
(19)$$\varphi_{n}^{s}\left({\tilde{p}}_{n}^{s}\right)=\{\begin{array}{c} {log}_{2}{\left({\left({\tilde{p}}_{1}^{s}+{\tilde{\sigma }}_{\beta_{s}\left(1\right)}^{s}\right)}^{\alpha_{\beta_{s}\left(1\right)}}\right)}^{B_{S}},n=1\\ {log}_{2}{\left(\left(\frac{{\left({\tilde{p}}_{n}^{s}+{\tilde{\sigma }}_{\beta_{s}\left(n\right)}^{s}\right)}^{\alpha_{\beta_{s}\left(n\right)}}}{{\left({\tilde{p}}_{n}^{s}+{\tilde{\sigma }}_{\beta_{s}\left(n-1\right)}^{s}\right)}^{\alpha_{\beta_{s}\left(n-1\right)}}}\right)\right)}^{B_{S}},n>1 \end{array}.$$

A detailed description of the equivalent transformation process from Equation (7) to Equation (12) is presented in Appendix A.

## 4. Single-Subchannel Resource Optimization

This section investigates the resource allocation problem on resource block s∈S with the upper limit of power budget $p^{s}$. As a fundamental module for multi-resource-block joint resource allocation solutions, our improved method contributes to designing more efficient optimization algorithms. The following equation can describe the single-resource-block optimization issue:(20)$$W_{RB}^{s}:W_{s}\left(p^{s}\right)=max\sum_{n=1}^{T}\varphi_{n}^{s}\left({\tilde{p}}_{n}^{s}\right)+C^{s}.$$
(21)$$w1^{*},w4^{*},w5^{*}:0\leq {\tilde{p}}_{T}^{s}\leq {\tilde{p}}_{T-1}^{s}\leq \ldots \leq {\tilde{p}}_{1}^{s}\leq P_{max}^{s};$$
(22)$$w3^{*}:\left|A_{s}*\right|\leq A.$$
On a single resource block, the constant term of the objective function is
(23)$$C^{s}={log}_{2}{\left(\frac{1}{{\tilde{\sigma }}_{\beta_{s}\left(T\right)}^{s}}\right)}^{\alpha_{\beta_{s}\left(T\right)}}.$$

## 4.1. Properties Analysis of the Functions $\varphi_{m}^{s}$

For *m* > *n*, we define a set of continuous variables ${\tilde{p}}_{n}^{s}={\tilde{p}}_{n+1}^{s}=\cdots ={\tilde{p}}_{m}^{s}$ as $\tilde{p}\in \left[0,p^{s}\right]$. An auxiliary function is proposed to assist us in analyzing $\varphi_{m}^{s}$:(24)$$\varphi_{n}^{s}\left(\tilde{p}\right)≜\sum_{l=n}^{m}\varphi_{l}^{s}\tilde{p}=\{\begin{array}{c} {log}_{2}{\left({\left({\tilde{p}}_{1}^{s}+{\tilde{\sigma }}_{\beta_{s}\left(1\right)}^{s}\right)}^{\alpha_{\beta_{s}\left(1\right)}}\right)}^{B_{S}},n=1\\ {log}_{2}{\left(\left(\frac{{\left({\tilde{p}}_{n}^{s}+{\tilde{\sigma }}_{\beta_{s}\left(n\right)}^{s}\right)}^{\alpha_{\beta_{s}\left(n\right)}}}{{\left({\tilde{p}}_{n}^{s}+{\tilde{\sigma }}_{\beta_{s}\left(n-1\right)}^{s}\right)}^{\alpha_{\beta_{s}\left(n-1\right)}}}\right)\right)}^{B_{S}},n>1 \end{array}.$$

Assuming that neither user *n* nor user (*m* − 1) occupies resource block *s*, these invalid users cannot allocate power from resource block *s*: ${\tilde{p}}_{n}^{s}={\tilde{p}}_{n+1}^{s}=\cdots ={\tilde{p}}_{m}^{s}$. Therefore, function $\varphi_{n,m}^{s}$ can supersede the value of $\sum_{l=n}^{m}\varphi_{l}^{s}$. Suppose that the upper limit for the number of valid users satisfies Condition *w*3*. In that case, each resource block can be allocated to at most *A* maritime users for multiplexing. Therefore, evaluating the calculative complexity of the target function in $W_{RB}^{s}$ requires the execution of *O*(*A*) operations.

Through the above simplification, the original optimization objective is transferred to the maximization of function $\varphi_{n,m}^{s}$ with constraint conditions. We solve the extreme points and inflection points of the function by analyzing the first and second derivatives of $\varphi_{n,m}^{s}$. Suppose that s∈S, m∈{1,2,…,T}, and *m* > *n*:
If *n* = 1 or the weight value satisfies $\alpha_{\beta_{s}\left(m\right)}\geq \alpha_{\beta_{s}\left(n-1\right)}$, $\varphi_{n,m}^{s}$ is monotonically increasing with convex in the interval [0,∞);If both the inequation of $\alpha_{\beta_{s}\left(m\right)}<\alpha_{\beta_{s}\left(n-1\right)}$ and *n* > 1 are satisfied, $\varphi_{n,m}^{s}$ shows a single peak. In this case, *C*_1_ refers to the extreme point. Function $\varphi_{n,m}^{s}$ increases in interval $\left(-{\tilde{\sigma }}_{\beta_{s}\left(n-1\right)},C_{1}\right]$ but decreases in interval $\left[C_{1},+\infty \right)$. Let *C*_2_ denote the inflection point. $\varphi_{n,m}^{s}$ is convex on $\left(-{\tilde{\sigma }}_{\beta_{s}\left(n-1\right)},C_{2}\right]$ and concave on $\left[C_{2},+\infty \right)$, where $C_{1}\leq C_{2}$:(25)$$C_{1}=\frac{\alpha_{\beta_{s}\left(n-1\right)}{\tilde{\sigma }}_{\beta_{s}\left(m\right)}-\alpha_{\beta_{s}\left(m\right)}{\tilde{\sigma }}_{\beta_{s}\left(n-1\right)}}{\alpha_{\beta_{s}\left(m\right)}-\alpha_{\beta_{s}\left(n-1\right)}},$$
(26)$$C_{2}=\frac{\sqrt{\alpha_{\beta_{s}\left(n-1\right)}}{\tilde{\sigma }}_{\beta_{s}\left(m\right)}-\sqrt{\alpha_{\beta_{s}\left(m\right)}}{\tilde{\sigma }}_{\beta_{s}\left(n-1\right)}}{\sqrt{\alpha_{\beta_{s}\left(m\right)}}-\sqrt{\alpha_{\beta_{s}\left(n-1\right)}}}.$$


The pseudo-code provided in Algorithm 1 expresses the process of calculating argmaxφn,ms(p˜), aiming to find the corresponding solution p˜∈[0,ps] when function φn,ms obtains the optimal value. The complexity of Algorithm 1 is *O*(1), since the calculation involves only a simple fixed number of fundamental operations. The works in this subsection facilitate the subsequent optimization for power and user allocation.
**Algorithm 1.** Compute argmaxφn,ms(p˜),p˜∈[0,ps]
 1:  **Input:** n,m,α,T,BS,Gts(t∈T),σts(t∈T),ps 2:  **Function** argmaxφ3:  x←βs(m)4:  y←βs(n−1)5:  **If** n=1 or αβs(m)≥αβs(n−1) **then**6:    **Return** ps7:  **Else**8:    **Return** max{0,min{αyσ˜x−αxσ˜yαx−αy,ps}}9:  **End if**10: **End Function**

### 4.2. Single-Resource-Block Power Allocation

First, we optimize the power allocation on a single resource block. When the maritime user allocation $A_{s}*$ on the resource block *s* is fixed, the optimizing problem is organized as follows:(27)$$W_{RBPA}^{s}\left(A_{s}*,p^{s}\right)=max\sum_{m=1}^{T}\varphi_{m}^{s}\left({\tilde{p}}_{m}^{s}\right)+C^{s}.$$
(28)$$w1^{*},w4^{*}:0\leq {\tilde{p}}_{T}^{s}\leq {\tilde{p}}_{T-1}^{s}\leq \ldots \leq {\tilde{p}}_{1}^{s}\leq P^{s}.$$

The maritime user $t\notin A_{s}*$ cannot obtain the allocated power without reusing the resource block *s*. Therefore, we exclude these invalid offshore users while calculating problem $W_{RBPA}^{s}$. The valid offshore users can be expressed as $m_{s}\in \left\{1_{s},2_{s},\cdots {\left|A_{s}*\right|}_{s}\right\}$, where *m* represents the SIC decoding order. For example, $A_{s}*=\left\{2,5,7,9\right\}$ indicates that there are four valid offshore users occupying resource block *s*: $1_{s}=2$, $2_{s}=5$, $3_{s}=7$, and $4_{s}=9$. To facilitate the statistical analysis, we add an index $0_{s}=0$. According to the definition, for m≥1, we can conclude that ${\tilde{p}}_{1}={\tilde{p}}_{2}>{\tilde{p}}_{3}={\tilde{p}}_{4}={\tilde{p}}_{5}>{\tilde{p}}_{6}={\tilde{p}}_{7}={\tilde{p}}_{8}={\tilde{p}}_{9}$. This equation also explains ${\tilde{p}}_{{\left(m-1\right)}_{s}+1}=\cdots ={\tilde{p}}_{m_{s}}$. Therefore, we only need to consider effective users when calculating $W_{RBPA}^{s}$. We can further simplify the optimization objective to $W_{RBPA}^{s}*$:(29)$$W_{RBPA}^{s}*=max\sum_{m=1}^{\left|A_{s}*\right|}\varphi_{{\left(m-1\right)}_{s}+1,m_{s}}^{s}\left({\tilde{p}}_{m_{s}}^{s}\right)+D^{s},D^{s}=\varphi_{|A_{s}*{|}_{s}+1,T}^{s}\left(0\right)+C^{s}.$$

For 1≤n≤m≤T, we can obtain $\varphi_{n,m}^{s}≜\varphi_{{\left(n-1\right)}_{s}+1,m_{s}}^{s}\left(A_{s}*\right)$. Then, we derive $argmax\varphi_{n,m}^{s}≜argmax\varphi_{{\left(n-1\right)}_{s}+1,m_{s}}^{s}\left(A_{s}*\right)$. Equation (29) can be transformed into the following formula:(30)$$W_{RBPA}^{s}*=max\sum_{m=1}^{\left|A_{s}*\right|}\varphi_{m,m}^{s}\left(A_{s}*,{\tilde{p}}_{m_{s}}^{s}\right)+D^{s}.$$
(31)$$w1^{*},w4^{*},w5^{*}:0\leq {\tilde{p}}_{T}^{s}\leq {\tilde{p}}_{T-1}^{s}\leq \ldots \leq {\tilde{p}}_{1}^{s}\leq P^{s}.\;$$

Algorithm 2 describes the power allocation algorithm on a single resource block (RBPA). In the subsequent multi-resource-block resource optimization, it is necessary to calculate the optimal power allocation on each resource block with various power budget values *P^s^*. As the RBPA program is repeatedly executed, the computational complexity will gradually increase. In lines 3–16 of Algorithm 2, we introduce Function 1 to calculate WRBPAs∗(As∗,ps). The results p˜1ss,p˜2ss,…,p˜|As∗|ss are stored in a lookup table. As shown in line 19, the final return values are limited by the power budget, since the optimized results have to meet the requirements of the constraint Ps≤Pmax.
**Algorithm 2.** Single-resource-block power allocation scheme (RBPA) 1:  **Input: α,T,BS,Gts(t∈T),σts(t∈T),As∗,Psmax** 2: ** Global variable: p˜1ss,p˜2ss…,p˜|As∗|ss**3:  **Function 1 α,T,BS,Gts(t∈T),σts(t∈T),As∗,Ps**4:  **For** n=1 **to** |As∗| **do**5:    #Calculate the optimum value of φm,ms in (30)6:    p˜∗←argmaxφ(n,n,α,T,BS,Gts(t∈T),σts(t∈T),As∗,ps) 7:    #Modify p˜∗ if this allocation dissatisfies condition w4∗8:    n←m−19:    **While** p˜nss<p˜∗ and n≥1 **do**10:    **p˜∗←argmaxφ(n,m,α,T,BS,Gts(t∈T),σts(t∈T),As∗,ps)**11:     n←n−112:   **End while**13:   p˜(n+)ss=⋯=p˜mss←p˜∗14: **End for**15: **Return** p˜1ss,p˜2ss…,p˜|As∗|ss16: **End Function 1**17: p˜1ss,p˜2ss…,p˜|As∗|ss←function1(α,T,BS,Gts(t∈T),σts(t∈T),As∗,Pmax
18: **Function 2** (ps)19: **Return** min{p˜1ss,ps},…,min{p˜|As∗|ss,ps}20: **End Function 2**

The maximum calculative complexity for power allocation is *O*(*A*^2^), and the subsequent evaluation work costs *O*(*A*). Suppose that there are *Q* different power budgets given in the multi-resource-block optimization. In that case, RBPA is used as the basic module for calculating the corresponding optimal power allocation with the complexity of *O*(*A*^2^
*+ QA*).

### 4.3. Single-Resource-Block Maritime User Allocation

Based on the previous subsection, we further optimize the maritime user allocation on a single resource block (RBUA) with the DP algorithm. Therefore, the effective user set $A_{s}*$ is a variable satisfying the upper limit in condition $w3*:\left|A_{s}*\right|\leq a,a\in \left\{0,1,\ldots ,A\right\}$. The main idea of the RBUA scheme is to recursively calculate the elements of three arrays: *ψ*, *X*, and *Y*. Let a∈{0,1,…,A}, n∈{1,2,…,T}, and m∈{n,n+1,…,T}. *ψ*[*a*, *n*, *m*] is defined as the optimal value for the following formula: *W_RB_*[*a*, *n*, *m*].
(32)$$W_{RB}\left[a,n,m\right]:\psi \left[a,n,m\right]=max\sum_{l=n}^{T}\varphi_{l}^{s}\left({\tilde{p}}_{l}^{s}\right).$$
(33)$$w1^{*}:{\tilde{p}}_{1}^{s}\leq P_{max}^{s};$$
(34)$$w3^{*}:\left|A_{s}*\right|\leq a;$$
(35)$$w4^{*}:{\tilde{p}}_{m}^{s}\geq {\tilde{p}}_{m+1}^{s};\;$$
(36)$$w5^{*}:{\tilde{p}}_{T+1}^{s}=0;$$
(37)$$w6^{*}:{\tilde{p}}_{n}^{s}={\tilde{p}}_{n+1}^{s}=\cdots ={\tilde{p}}_{m}^{s}.$$

The array *Y* is responsible for recording the previous elements contributing to the current optimal value *ψ*[*a*, *n*, *m*]. In Equation (32), it is observed that the target function depends only on ${\tilde{p}}_{n}^{s},{\tilde{p}}_{n+1}^{s},\ldots ,{\tilde{p}}_{T}^{s}$. If a=A,n=m=1 is true, array *ψ*[*A*, 1, 1] is the best value for the problem $W_{RB}^{s}$. When *W_RB_*[*a*, *n*, *m*] achieves the maximum value, we define ${\tilde{p}}_{n}^{s}*,{\tilde{p}}_{n+1}^{s}*,\ldots ,{\tilde{p}}_{T}^{s}*$ as the optimal solution corresponding to *ψ*[*a*, *n*, *m*]. Let array X[a,n,m] record the corresponding solution of the optimal objective function. Then, we combine Condition *w*6* to derive $X\left[a,n,m\right]={\tilde{p}}_{n}^{s}*={\tilde{p}}_{n+1}^{s}*=\cdots ={\tilde{p}}_{m-1}^{s}*={\tilde{p}}_{m}^{s}*$. Based on the relation in Equation (38), the implementation of the marine user allocation is a recursive process used to calculate the elements of ***ψ***.
(38)$$\psi \left[a,v,m\right]=\{\begin{array}{c} \varphi_{1},\;\varphi_{1}>\varphi_{2},\tilde{p}*>X\left[a-1,m+1,m+1\right]\\ \varphi_{2},Otherwise \end{array},$$
(39)$$\varphi_{1}=\varphi_{n,m}^{s}\left(\tilde{p}\right)+\psi \left[a-1,m+1,m+1\right],$$
(40)$$\varphi_{2}=\psi \left[a,n,m+1\right].$$

In Equation (38), $\tilde{p}*=argmax\varphi \left(n,m,P^{s}\right)$ and *φ*_1_ records the corresponding optimal value ***ψ***[*a*, *n*, *m*] for a valid user *m*. On the contrary, *φ*_2_ represents the optimum value for an invalid user on resource block *s*.

During the iterative procedure shown in Algorithm 3, the array *Y* tracks and records the previous elements contributing to the current function *ψ*[*a*, *n*, *m*]. This can be traced back from index (*A*, 1, 1) to an empty set. This method assists us in retrieving the optimal solution for the entire program process more conveniently in lines 29–36. From lines 6–12, we initialize the index as ∅ to represent the end of recursion. In summary, the recursive relationship among these three arrays (*ψ*, *X*, *Y*) is divided into four cases:

Case 1 (line 4–6): If $\psi \left[a,n,m\right]=\varphi_{1}$,
(41)$$X\left[a,n,m\right]={\tilde{p}}^{*},$$
(42)Y[a,n,m]=(a−1,m+1,m+1).

Case 2 (line 10–12): If $\psi \left[a,n,m\right]=\varphi_{2}$,
(43)X[a,n,m]=X[a,n,m+1],
(44)Y[a,n,m]=(a,n,m+1).

Case 3 (line 20–22): If a=0, no valid user is reusing the resource block and
(45)$$\psi \left[0,n,m\right]=\varphi_{n,T}^{s}\left(0\right),$$
(46)X[0,n,m]=0,
(47)Y[0,n,m]=∅.

Case 4 (line 24–26): If m=T,n≤T,
(48)$$\psi \left[a,n,m\right]=\varphi_{n,T}^{s}\left({\tilde{p}}^{*}\right),$$
(49)$$X\left[a,n,m\right]={\tilde{p}}^{*},$$
(50)Y[a,n,m]=∅.

**Algorithm 3.** Single-resource-block maritime user allocation scheme (RBUA) 1:  **Function RBUA** (α,T,BS,Gts(t∈T),σts(t∈T),A,ps)  2:  **#Initialize arrays**
***ψ***, ***X***, ***Y*** for a=0,m=T3:  **For m=T** to 1 and n=m to 1 **do**4:    ψ[0,n,m]←φn,Ts(0)5:    X[0,n,m]←06:    Y[0,n,m]←∅7:  **End for**8:  **For a=1** to A and n=T to 1 **do**9:  p˜∗←argmaxφ(n,T,α,T,BS,Gts(t∈T),σts(t∈T),As∗,ps)10:    **ψ[0,n,m]=φn,Ts(p˜∗)**11:    X[0,n,m]=p˜∗12:    Y[0,n,m]=∅13: **End for**14: #Calculate ψ,X,Y for a∈[1,A] and n≤m≤T−115: **For m=T−1** to 1 and a=1 to *A* and n=m to 1 **do**16:   p˜∗←argmaxφ(n,m,α,T,BS,Gts(t∈T),σts(t∈T),ps)17:   φ1←φn,ms(p˜∗)+ψ[a−1,m+1,m+1]18:   φ2←ψ[a,n,m+1]19:   **If** φ1>φ2 and p˜∗>X[a−1,m+1,m+1]
**then** 20:     ψ[a,n,m]←φ121:     X[a,n,m]←p˜∗22:     Y[a,n,m]←(a−1,m+1,m+1)23:   **Else**24:     ψ[a,n,m]←φ225:     X[a,n,m]←X[a,n,m+1]26:     Y[a,n,m]←(a,n,m+1) 27:   **End if**28: **End for**29: #Retrieve the optimum solution p˜S30: p˜1s,…,p˜Ts←031: (a,n,m)←(A,1,1)32: **While** (a,m,m)≠∅ **do**33:   p˜ns,…,p˜ms←X[a,n,m]34:   (a,n.m)←Y[a,n,m]35: **End while**36: **Return P˜S**37: **End Function RBUA**

To avoid the repeated execution of the DP procedure in the multi-resource-blocks system, Algorithm 4 implements a pre-calculated improvement strategy. This subsection essentially optimizes user allocation based on power allocation in the previous subsection. Hence, we define the improved algorithm as a joint power and user allocation scheme on a resource block (RBPUA) to distinguish it from RBUA. In lines 1–4, the algorithm executes RBPA to obtain three arrays (*ψ*, *X*, *Y*). From lines 5–8, we solve the valid user set As∗ in *ψ*[*A*, 1, *m*] and the corresponding optimal solution p˜1,p˜2,….,p˜T, then store these values in the global variable set. In line 10, the objective function WRBPAs(As∗,ps) is maximized by searching for the best maritime user selection among *T* possibilities in the set. At the end of the algorithm, the returned result is restricted by the power budget *p^s^*(*P^s^* ≤ *P*_max_) in RBPA.
**Algorithm 4.** Improved scheme for single-resource-block power and user allocation (RBPUA) 1:  **Input:** α,T,BS,Gts(t∈T),σts(t∈T),As∗,Pmax 2:  **Global variable:** G3:  **Initialization:**4:  From RBUA(α,T,BS,Gts(t∈T),σts(t∈T),As∗,Pmax to obtain ***ψ***, ***X***, ***Y***
5:  **For** m=1 to *T*
**do**6:   Retrieve the effective users set As∗ of ψ[A,1,m] and the corresponding optimum solution p˜1s,p˜2s,…,p˜Ts7:   Add (As∗,p˜1s,p˜2s,…,p˜Ts) to G8:  **End for**9:  **Function RBPUA(ps)**10: Get (As∗,p˜1s,…,p˜Ts) which maximizing WRBPAs(As∗,ps)=∑l=1|As∗|φl,ls(As∗,min{p˜lss,ps})+Ds from G11: **Return** min{p˜1ss,ps},…,min{p˜Ts,ps}12: **End Function RBPUA**

The improved RBPUA scheme is prepared for the efficient calculation of the multi-resource-block optimization problem. Consequently, on a single resource block with a power budget, the calculative complexity achieved by both RBPUA and RBUA is equal to *O*(*AT*^2^). When calculating the optimal $W_{RB}^{s}$ corresponding to *Q* different power budgets in a multi-resource-block problem, the RBPUA with a computational complexity of *O*(*AT*^2^ + *QAT*) is more efficient than RBUA with *O*(*QAT*^2^). We can make an intuitive comparison from the numerical simulation given in Section 6.

## 5. Multi-Subchannel Resource Optimization

Based on the single-resource-block optimization, the multi-resource-block joint maritime user and power allocation problem can be defined by the following formula:(51)$$W_{MRB}=max(\sum_{s\in S}W_{s}\left(p^{s}\right)).$$
(52)$$C1:W_{s}\left(p^{s}\right)=max\sum_{n=1}^{T}\varphi_{n}^{s}\left({\tilde{p}}_{n}^{s}\right)+C^{s};$$
(53)$$C2:\sum_{s\in S}^{\sum^{\;}}P^{s}\leq P_{max};$$
(54)$$C3:0\leq P^{s}\leq P_{max}^{s}.$$

For power budget *P^s^*, the best resource allocation result can be obtained quickly for each resource by invoking the RBPUA scheme. Therefore, the power budget directly impacts the optimization solution of power and marine user allocation. *W_MRB_* is ultimately the process of optimizing these two variables (${\tilde{p}}^{s}$ and $A_{s}*$) with *P^s^* together. Suppose we can replace several definitions in *W_MRB_* with $W_{s}\left(p^{s}\right)$ and conditions in problem $W_{RB}^{s}$. In that case, *W_MRB_* will be transformed into the equivalent problem *W** put forward at the beginning of this paper.

### 5.1. Heuristic Optimization Algorithm Based on Gradient Descent (GRAD)

The improved gradient descent algorithm combined with the RBPUA can effectively solve the joint power and sea user allocation problems. Hence, this paper regards the heuristic algorithm as the performance standard of the multi-resource-block joint optimization scheme. Compared with the traditional optimization algorithm, this scheme offers the advantage of reducing the complexity and achieving the optimal WAR. Its principle lies in the optimization of two stages. Specifically, the first stage follows the projected gradient descent of the power vector $P^{s}=\left(P^{1},P^{2},\ldots ,P^{S}\right)$ in the search space formed by constraints (problem $W_{MRB}$: C2 and C3).

The second stage calculates *W_s_*(*p^s^*) as well as its first derivative value $W_{s}{}^{\prime }\left(p^{s}\right)$. Then, these two values return to the first stage and participate in the judgment of the iteration conditions. After this, we select an appropriate step size to decide which point to start the next iteration. The step length *λ* is determined by the optimization of *W_s_* along the ray $\left\{P^{s}+\lambda △|\lambda \geq 0\right\}$. We adopt the precise straight-line search method in this paper.
(55)$$\lambda =\underset{x\geq 0}{\mathrm{argmin}}W_{s}(\left(P^{s}+x△\right).$$

Let $P_{1}^{S},P_{2}^{S},\ldots ,P_{T}^{S}$ represent the output power allocation value of the RBPUA algorithm. The left derivative of $W_{s}{}^{\prime }\left(p^{s}\right)$ concerning *p^s^* can be described as follows [32]:(56)$$W_{s}{}^{\prime }\left(p^{s}\right)=\frac{B_{s}\alpha_{\beta_{s}\left(l\right)}}{\left({\tilde{p}}_{l}^{s}+{\tilde{\sigma }}_{\beta_{s}\left(l\right)}^{s}\right)ln2}=\frac{B_{s}\alpha_{\beta_{s}\left(l\right)}}{\left(P^{s}+{\tilde{\sigma }}_{\beta_{s}\left(l\right)}^{s}\right)ln2},$$
where *l* represents the index value when ${\tilde{p}}_{l}^{s}$ reaches the upper limit of the power budget on this resource block *s*.

Algorithm 5 describes the pseudo-code of the GRAD algorithm. In the third line, we initialize the fault tolerance *ξ* to represent the iteration termination condition. From lines 5 to 6, the gradient of ∑s∈SWs evaluated at the power budget vector Ps denotes the search direction. At line 7, we can adjust the *λ* by means of a precise straight-line search simulation. Finally, the projection of Ps+λ△ in the search constraint space is calculated effectively at line 8.
**Algorithm 5.** Heuristic algorithm based on gradient descent (GRAD) 1:  **Function GRAD (Input:α,T,BS,Gts(t∈T),σts(t∈T),A,Pmaxmaxs**)  2:  Let Ps←0 be the starting point3:  **While ‖Ps∗−Ps‖>ξ**
**do**4:   Conserve the preceding vector Ps∗←Ps5:   Determine a search direction △6:   △←(W1′(p1),W2′(p2),….,WS′(pS))7:   Select a step length λ8:   Update Ps← projection of Ps+λ△ in search space: feasible set {Ps:C2:∑s∈S∑ smaxsPs≤Pmax 9:  **End while**10: **Return** Ps11: **End Function GRAD**

When we adopt RBUA as the basic module in the multi-resource-block joint optimization, the GRAD algorithm achieves the complexity of $O\left(log(1/\xi \right)SAT^{2})$. However, the application of our improved method RBPUA to the GRAD scheme reduces the complexity to $O\left(SAT^{2}+log(1/\xi \right)SAT)$.

### 5.2. Multiple-Choice Knapsack Algorithm Based on Dynamic Programming (MCKP-DP)

We have simplified and transformed the objective function before to deal with the problems of joint marine users and power allocation more simply. This process depends on the change in variable *p^s^* with a continuous search space. However, the research on NP-hard optimization and the relevant approximate algorithm requires that the accuracy of the parameters and variables be bounded. In that case, the discretization of variables can be adopted to simplify the complexity [33]. This method of discrete simplification is suitable for practical systems limited by a minimum transmit power at the coastal BS. In conclusion, we can discretize the seek constraint space similarly to the method in reference [9]. We set the minimum discrete transmit power value as *p*. The power budget *p^s^* on each resource block is a multiple of the minimum power *p*: $P^{s}=j\times p,j\in \left\{0,1,2,..,\lfloor \frac{P^{s}}{p}\rfloor \right\}$. Let $J=\lfloor \frac{P^{s}}{p}\rfloor$ represent the total number of items with an allocated power. Then, we discretize the constraint conditions of the optimization function into a feasible set *F*, which is expressed as follows:(57)$$F=\{P^{s}:\sum_{s\in S}P^{s}\leq P_{\max},0\leq P^{s}\leq P_{\max}^{s},s\in S,P^{s}=j\cdot p,j\in \{0,1,2,..,J\}\}$$

Through the discretization of the power budget on each resource block, problem *W_MRB_* can be transformed into the classic multiple-choice knapsack problem *W_MCKP_* [28]:(58)$$W_{MCKP}=max\sum_{s\in S}\sum_{j=1}^{J}pro_{j}^{s}z_{j}^{s}.$$
(59)$$F1:\sum_{s\in S}\sum_{j=1}^{J}wei_{j}^{s}z_{j}^{s}\leq P_{max};$$
(60)$$F2:\sum_{j=1}^{J}wei_{j}^{s}z_{j}^{s}\leq P_{max}^{s};$$
(61)$$F3:\sum_{j=1}^{J}z_{j}^{s}\leq 1;$$
(62)$$F4:z_{j}^{s}\in \left\{0,1\right\},j\in \left[1,J\right],s\in S.$$

In the *W_MCKP_* problem, we define *S* non-intersecting classes to represent *S* non-interfering resource blocks. Each class includes *J* items packed into a backpack with a capacity of *P*_max_. The profit of each project $pro_{l}^{s}$ refers to the WAR value of item *l* on resource block *s*. Moreover, the weight $wei_{l}^{s}$ represents the power allocated to item *l*. The knapsack problem stipulates that only one item from each category is selected into the backpack to maximize the total profits. At the same time, the weights are constrained by the $P_{max}^{s}$ of each category and the maximum capacity *P*_max_ of the backpack. Therefore, when the item *j* on resource block *s* is allocated to the knapsack, binary variable $z_{l}^{s}=1$. Finally, *W_MCKP_* is able to approach the continuous solution of GRAD with an arbitrary accuracy by adjusting *p*.

The error between the optimal result $W_{MCKP}*$ of the discrete scheme and the optimal result $W_{MRB}*$ of the continuous algorithm is as follows:(63)$$W_{MRB}*-W_{MCKP}*\leq \delta \sum_{s\in S}\underset{t\in T}{max}\left\{\frac{B_{s}\alpha_{\beta_{s}\left(t\right)}}{\left(P_{s}^{*}+{\tilde{\sigma }}_{\beta_{s}\left(t\right)}^{s}\right)ln2}\right\},$$
where $P_{s}^{*}$ refers to the optimal power budget on resource block *s* in problem *W_MRB_*. The discretization error depends on the upper limit of a linear function in *δ* with the coefficient decided by the system parameters. Appendix B gives the detailed derivation process of Equation (63).

*W_MCKP_* in polynomial time can be dealt with by DP, which is summarized as weighted DP and profit DP [28]. Since the MCKP is able to dynamically solve the maximum profit by discretizing weights, we adopt the principle of weight-based DP. Let *Z* be a two-dimensional array. *Z*[*s*, *j*] is defined as the optimal value of *W_MCKP_* limited to the first *s* categories with a capacity of *j* × *p*. The recursive relationship for ***Z*** on resource block *s* is as follows:(64)$$Z\left[s,j\right]=\underset{l\leq j}{max}\left\{Z\left[s-1,j-l\right]+pro_{l}^{s}\right\}.$$

As shown in Algorithm 6, through the application of the weight-based DP idea to the multiple-choice knapsack algorithm, we propose the use of the MCKP-DP scheme to handle *W_MCKP_*. First of all, we convert the continuous joint optimization problem *W_RB_* to the discrete search space problem *W_MCKP_*. Locating lines 6 and 7, the RBPUA algorithm calculates the profit $pro_{i}^{s}$ for each project. In line 8, we perform weight-based DP.
**Algorithm 6.** Multiple-choice knapsack algorithm based on dynamic programming (MCKP-DP) 1:  **$\mathrm{Function}\;\text{MCKP-DP}\;(\alpha ,T,B_{S},G_{t}^{s}\left(t\in T\right),\sigma_{t}^{s}\left(t\in T\right),A,P_{{max}_{max}}^{s}$**
 2:  #Calculate the parameters of MCKP3:  **For s∈S**
**do**4:    **For j∈[0,J]**
**do**
5:      **Forl∈[0,j]**
**do**
6:        prols←l⋅p7:        weils←WRBPUAs(l⋅p)8:        Zs[s,j]←max{Z[s−1,j−l]+prols} 9:      **End for**10:   **End for**11: **End for**12: Backtracking to obtain the power vector corresponding to Zs[s,j]13: **Return** the optimal allocation by means of weights-based dynamic programming [28]14: **End Function MCKP-DP**

### 5.3. Fully Polynomial Time Approximation Program Based on Dynamic Programming (DP-FPTA)

Aiming at an NP-hard optimization conundrum, the knapsack algorithm and related approximation schemes can effectively ensure that the approximation precision reaches any degree required. Let Π be an NP-hard optimization problem with objective function *f*_Π_; we propose an approximation scheme for the optimization problem on the input (*I*, *ε*), where *I* represents the instance of Π and *ε* > 0 is taken as the error parameter. The optimal result of function *f*_Π_(*I*, *ε*) satisfies the requirements of Equation (65):(65)$$\{\begin{array}{c} f_{\prod^{\;}}\left(I,\varepsilon \right)\leq \left(1+\varepsilon \right)\cdot Optimal,\left(\prod^{\;}is\;a\;minimization\;problem\right)\\ f_{\prod^{\;}}\left(I,\varepsilon \right)\geq \left(1-\varepsilon \right)\cdot Optimal,\left(\prod^{\;}is\;a\;maximization\;problem\right) \end{array}.$$

This scheme is called the polynomial-time approximation (PTA) algorithm. For each fixed *ε* > 0, its running time is limited by a polynomial with the size of instance *I*. In the subsequent more rigorous approximation concept, FPTA limits the running time required by the polynomial with both instance size and 1/*ε*. From the technical perspective, FPTA is the most providential strategy to solve the NP-hard optimization problem.

To deal with the pseudo-polynomial complexity of *J*, we will formalize this concept and develop a complete polynomial approximation algorithm by combining it with dynamic programming. The core of the DP-FPTA algorithm is that for any *ε* > 0, the final optimization result is higher than the lower bound of performance guarantee: $\left(1-\varepsilon \right)\cdot W_{MCKP}*$. In addition, the execution time is limited by both the size of the polynomial input and 1/*ε*. Supposing that *P* ≠ *NP*, the DP-FPTA algorithm is the best compromise solution to the NP-hard optimization problem. The algorithm ensures the performance of WAR while reducing the computational difficulty [34].

As shown in Algorithm 7, the DP-FPTA algorithm is based on a combination of profit-based dynamic programming and an approximate optimal scheme. Scaling profit generally contributes to the simplification of the items calculated in the knapsack problem. We first give an estimate *F** of the optimal value of the *W_MCKP_* problem; the estimated value satisfies the requirements of the condition $F*\geq W_{MCKP}*\geq \frac{F*}{4}$ [34]. To further simplify the workload, we focus only on subset *J_s_* of the items for each category rather than calculating the profit value $pro_{j}^{s}$ of the total items for *S* categories.
(66)$$J_{s}=\{l\leq J,j\leq \frac{4S}{\varepsilon }-1,pro_{l-1}^{s}<j\frac{\varepsilon F*}{4S}\leq pro_{l}^{s}\}.$$

The discrete terms of the DP-FPTA scheme can be expressed as j∈{1,2,…,4Sε} to transform the continuous profit space into discrete values for every interval [(j−1)⋅εF∗4S,j⋅εF∗4S]. Subsequently, we introduce a multi-key binary search to obtain subset *J_s_* on each resource block [35]. Finally, as shown in lines 11–14 of Algorithm 7, we adopt profit-based DP to obtain the optimal solution for resource allocation. Let *Q* denote a DP array for the smallest weight of the subproblem composed of *S* categories (resource blocks), the problem of MCKP be limited to the first *s* categories, and *Q*(*s*,*q*) denote the minimum power budget required to achieve a WAR of *q*·*εF**/4*S*. After initializing *Q*[0,0] = 0, *Q*[0,*q*] = +∞ and letting q=0,1,…,⌊4Sε⌋, the recursive relationship can be expressed as
(67)[s,q]=minj∈Js{Q[s−1,q−⌊4projsSεF∗⌋]+weijs,q⋅εF∗4S≥projs+∞, Otherwise,
where |Js|=min{4Sε,J}. Furthermore, the gap between the DP-FPTA optimal solution and the MCKP-DP optimum WMCKP∗ is, at most, ε⋅WMCKP∗. The calculative complexity of DP-FTPA is O(SAT2+min{(log2J)S2ATε+S3ε2,JSAT+J2S}.
**Algorithm 7.** Fully polynomial time approximation program based on dynamic programming (DP-FPTA) 1:  **Function DP-FPTA** (α,T,BS,Gts(t∈T),σts(t∈T),A,Pmaxmaxs
 2:  Calculate an estimated value F∗ of WMCKP∗, F∗≥WMCKP∗≥F∗43:  **For s∈S**
**do**4:    K=εF∗/S
5:      **For j∈Js**
**do**6:        **While proj+1s−projs>K**
**do**7:        Obtain projs,weijs through multi-key binary search method 8:        **End while**9:      **End for**10: **End for**11: **For s∈S**
**do**12:   **For q∈[0,1,…,⌊4Sε⌋]**
**do**13:   Q[s,q]=min{Q[s−1,q−⌊4projsSεF∗⌋]+weijs}14:   **End for**15: **End for**16: **Return** DP-FPTA allocation by means of the profits-based dynamic programming [28] 17: **End Function DP-FPTA**

Remark: In [34], it is pointed out that in a very technical sense, the FPTA scheme is the best one can expect for an NP-hard optimization problem, assuming that P≠NP. The design of almost all the FPTA and PTA schemes is based on the idea of trading accuracy for running time. Inspired by the idea of an approximate algorithm, we can deal with the NP-hard problem in the following three ways. (1) If the input scale of the problem is small, the search strategy can be used to solve the problem in exponential time. If the input scale is large, (2) a random algorithm can accurately solve the problem with a high probability in polynomial time or (3) an approximate solution for the problem can be obtained in polynomial time. In [27], other approximate schemes are also mentioned. The central concept is to adopt the random sampling of sub-channels with probability *ρ* to avoid the traversal search. The simulation results show that, compared with the traditional optimization algorithm, the greedy-based approximate scheme saves 66% of the running time while maintaining the approximate optimal data rate. The question of whether the FPTA is the most desirable approximation algorithm will be explored in our future work.

## 6. Numerical Results and Discussion

In this paper, the model scenario simulated is located within the region of the *South China Sea*. Therefore, the *tropical ocean climate* is taken for the setting of the climate parameter. The coastal 5G BS serves an offshore area with a radius of 5 km, and there are *T* maritime users (ships or offshore reef islands) randomly distributed across the sea. The total channel bandwidth *B* of the offshore communication system is 5 MHz. Moreover, the system channel is allocated to 10 resource blocks with a bandwidth of *B_s_* = *B*/*S* = 0.5 MHz. When there is no typhoon, the sea surface is relatively calm, with the irregular terrain parameter taken as 0 [29]. Table 1 shows the reference values of the terrain irregularity parameter Δ*h*. The coastal BS is located in a high-lying area with a broad view. Therefore, its deployment parameter is *very careful*. In contrast, the deployment parameters of maritime users are relatively *random*. The effective heights of the BS transmitting antenna and the maritime user receiving antenna are 15 m and 5 m, respectively. In the multiple-choice knapsack program, we set the minimum discrete power to 0.01 W. Table 4 shows more detailed simulation parameters.

First, we test the path loss of the Longley–Rice model with the increase in communication distance under an offshore background. Our objective is to investigate the influence of these two parameters on the maritime channel model, including BS transmitting antennas with different heights *H_T_* and different degrees of irregularity ∆*h*. After that, we test the performance optimization and complexity of the proposed algorithm RBPUA based on NOMA and traditional OMA, respectively. Furthermore, we evaluate the three multi-resource-block joint optimization schemes not only in terms of the WAR performance optimization results but also in terms of the calculative complexity.

When the height of the BS antenna varies from 5 to 25 m, Figure 2 shows that the Longley–Rice transmission loss results in the increase in the transmission distance. Specifically, within 5 km, the transmission loss of the 5 m transmitting antenna is significantly greater than that of the antenna transmitting over 10 m. At a distance of 5 km, the loss values of the 10 m to 25 m antennas are almost equal to 114.7 dB. With the expansion of the communication range, the transmission loss corresponding to the 10 m BS antenna will increase instantaneously. When the distance reaches 8 km, the 15 m antenna will expose the defect of limited coverage compared with the antenna above 20 m. Considering the limited coverage of the shore-based 5G BS, the communication range in this paper is within a radius of 5 km over the sea. Therefore, if the height of the 5G BS transmitting antenna is more than 10 m, a good transmission loss result can be maintained in our subsequent work.

Figure 3 shows the path loss value achieved with the increase in distance based on different terrain irregular parameters. We set the effective height of the coastal BS transmitting antenna as 15 m. Within the communication distance of 5 km specified in this paper, the propagation loss values corresponding to irregular parameters Δ*h* = 0 and Δ*h* = 5 are almost equal. Referring to previous test results, we can regard the sea surface as calm and smooth under an offshore background without typhoon invasion. Therefore, in the subsequent simulation process, we suppose that the terrain irregularity parameter is zero. This value is still applicable even if there are slight waves on the sea surface.

The transmission loss reflected by the experimental results (Figure 2 and Figure 3) is the absolute value. In the subsequent simulation, those propagation loss values are negative in the maritime communication system.

Figure 4 compares the WAR performance optimization between RBUA and the improved scheme RBPUA based on NOMA and OMA, respectively. The NOMA system supports an upper limit of 10 marine users multiplexed on a resource block. At the same time, the power budget provided by the coastal BS increases from 5 W to 50 W. When the power budget reaches 10 W, the WAR performance of the NOMA system with a single resource block is increased by 4.08% compared with OMA. As the power budget increases to 50 W, the WAR of NOMA becomes better than that of OMA by 4.59%. In addition, we prove that the low-complexity RBPUA scheme is the same in terms of performance optimization as the RBUA scheme.

For the NOMA or OMA system with 10 power budgets, Figure 5 compares the computational complexity for the RBUA algorithm and the improved optimization algorithm RBPUA. According to the numerical result, for 10 maritime users, the NOMA-based RBPUA scheme is less complex than the NOMA-based RBUA algorithm by 80%. Consequently, applying our improved algorithm RBPUA in the multi-resource-block schemes can effectively reduce the computational difficulty while maintaining the performance optimization results.

In Figure 6, we compare the WAR optimization performance of three multi-resource-block joint power and maritime user allocation schemes (GRAD heuristic algorithm, MCKP-DP scheme, and DP-FPTA approximation problem). The offshore communication system serves 80 offshore users, including 10 resource blocks. Furthermore, the total power budget provided by BS increases from 5 W to 50 W. It can be seen from the simulation result that the GRAD with an accuracy of 0.0001 can be defined as the reference standard for optimal WAR. At the same time, the MCKP-DP scheme (the number of discrete items *J* = 1000) is comparable to the optimization performance of the GRAD program. When the coastal BS provides a power budget of 50 W, the WAR of the above two schemes in the NOMA system is increased by 4.53% compared with the OMA performance. For a NOMA maritime system with a 50 W power budget, the performance of our proposed DP-FPTA with a 0.01 approximate accuracy is 1.11% higher than that of the DP-FPTA with a 0.1 accuracy. In particular, the DP-FPTA scheme with an error of 0.01 almost achieves the optimized performance of GRAD and MCKP-DP.

Furthermore, Figure 7 exhibits the corresponding histogram for convenience; therefore, we can clearly distinguish the WAR simulation results for each optimization scheme.

Figure 8 provides the histograms for the numerical changes of WAR as the number of maritime users increases from 10 to 80. The simulation result shows that our communication system obtains the maximum WAR with 50 users. When maritime users continue to increase beyond this saturation state, the heavy load of channel reuse worsens user interference, resulting in a decline in system data rates. Moreover, in the saturation state, both GRAD and MCKP-DP solutions (number of discrete items: *J* = 1000) achieve an almost equivalent optimization performance of 9.421 × 10^7^ bit/s. Their WAR optimization results are higher than those of OMA by 7.47%. In this saturated NOMA system, the DP-FPTA with an 0.01 approximate accuracy achieves a WAR performance of 9.420 × 10^7^ bit/s comparable to that of GRAD and MCKP-DP. Meanwhile, it promotes the WAR by 1% over the approximate scheme with an accuracy of 0.1.

Figure 9 displays the optimization result of WAR for the DP-FPTA algorithm versus the increase of 4*S*/*ε*. Taking the optimization performance of the MCKP-DP scheme as a reference and *J* = 1000 as the total number of discrete items on each resource block, we normalize the *x*-axis by |*J_s_*| = 4*S*/*ε*. Evaluating the subset *Js* of projects on each resource block makes it more convenient to simulate the relationship between WAR and approximation accuracy *ε*. Our system, equipped with 10 resource blocks and a power budget of 10 W, serves 80 maritime users. In Figure 9, we also propose that the minimum performance index of the DP-FPTA is at least $\left(1-\varepsilon \right)\cdot W_{MCKP}*$. The result indicates that the WAR is higher than the performance index in both OMA and NOMA marine systems. As the number of items increases, the DP-FPTA-optimized performance gradually approaches the MCKP-DP performance curve. Moreover, when 4*S*/*ε* = 500 (*ε* = 0.08), the WAR of our approximate scheme will reach 99.55% of the MCKP-DP scheme.

Figure 10 further compares the calculative complexity of DP-FPTA and MCKP-DP. The complexity of the DP-FPTA scheme increases with the increase in 4*S*/*ε*. When the number of normalized items reaches 2900 (*ε* ≤ 0.01), its complexity is equivalent to that of the MCKP-DP scheme (*J* = 1000), and MCKP-DP achieves the optimal WAR. In this case, we do not recommend our proposed DP-FPTA scheme. Combined with the simulation results shown in Figure 9, within the approximate error interval of 0.01 ≤ *ε* ≤ 0.08, NOMA-based DP-FPTA achieves an approximately optimal WAR performance with far less calculative complexity than MCKP-DP. When the error parameter satisfies *ε* = 0.08, the WAR of the approximate scheme achieves 99.55% of MCKP-DP, but the complexity is 84.3% lower than the MCKP-DP scheme. Furthermore, the performance of the NOMA-based DP-FPTA is better than that of OMA by 4.48%. Results have shown that while controlling the computational complexity, DP-FPTA can ensure the optimization of WAR by adjusting the approximate error function *ε*. Therefore, our proposed scheme successfully achieves the trade-off between optimization performance and calculative delay.

## 7. Conclusions

This paper investigates the WAR optimization problem in the NOMA-based offshore downlink communication scenario of the *South China Sea*. We established a Longley–Rice-based maritime channel model, including various environmental factors and antenna deployment parameters. This model, which has adjustable environmental parameters, can be widely applied to various offshore communication applications around the world. Aiming at the single-resource-block resource allocation problem, we performed pre-computation to simplify the power and marine user allocation. Taking the improvement scheme RBPUA as the basic module, we developed three optimization solutions for multi-resource-block joint power and maritime user allocation. The simulation results indicate that we can regard the GRAD as a performance optimization index. MCKP-DP provides a new idea based on the knapsack problem for resource allocation. Its optimization performance achieved the best value, while the complexity had to be reduced. We proposed the use of the DP-FPTA to effectively weigh the relationship between computation complexity and performance optimization. This algorithm is suitable for a practical NOMA-based maritime communication system with a low time delay.

In subsequent work, the NOMA-based relay technology should be introduced into marine communications to expand the network coverage [36]. In addition, we will further expand the single-BS NOMA system to the maritime communication network jointly deployed by multiple offshore BSs. Moreover, motivated by the optimization schemes in this paper, we will delve into the problem of resource allocation in the MIMO-NOMA maritime system. The joint optimization project with beamforming and user clustering represents a significant challenge for our future work [37]. We aim to achieve improved signal strength and regional coverage while reducing signal interference in the next generation of this maritime communication system.

## Figures and Tables

**Figure 1 entropy-23-01454-f001:**
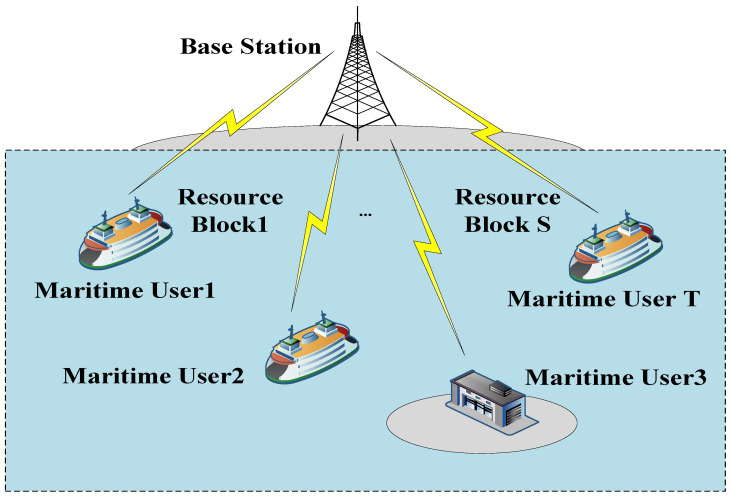
NOMA-based downlink maritime communication system.

**Figure 2 entropy-23-01454-f002:**
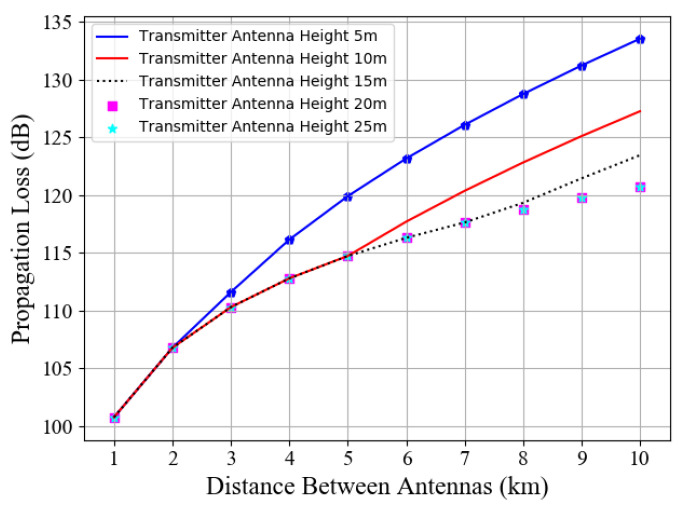
Longley–Rice transmission loss of different BS antenna heights vs. communication distances.

**Figure 3 entropy-23-01454-f003:**
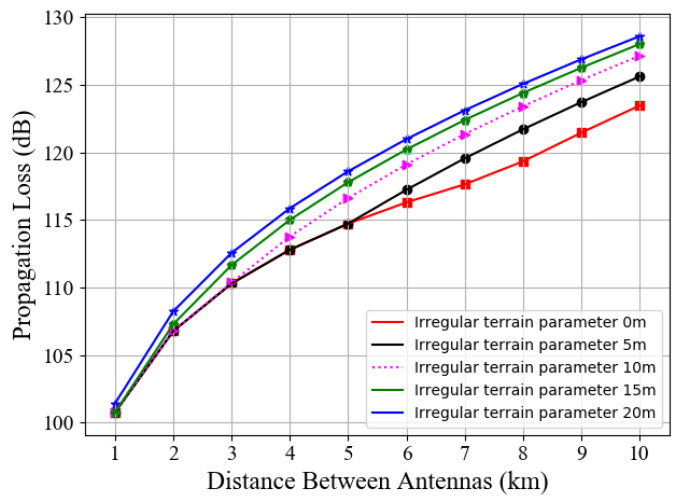
Longley–Rice transmission loss of different irregular terrain parameters vs. communication distances.

**Figure 4 entropy-23-01454-f004:**
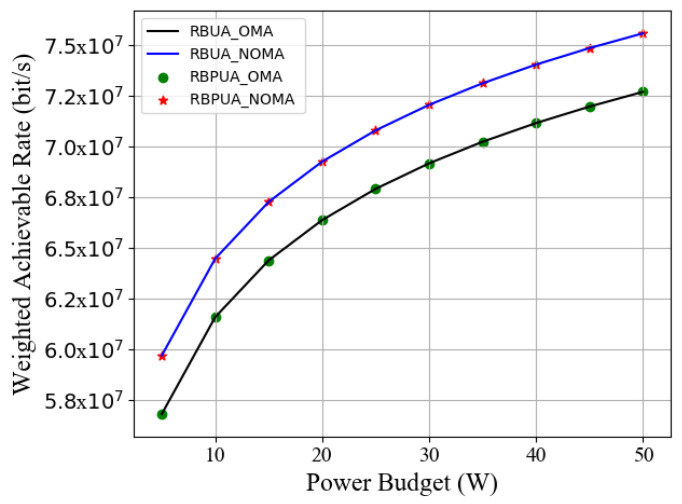
Weighted achievable rate of RBPUA and RBUA schemes vs. power budget in NOMA or OMA communications.

**Figure 5 entropy-23-01454-f005:**
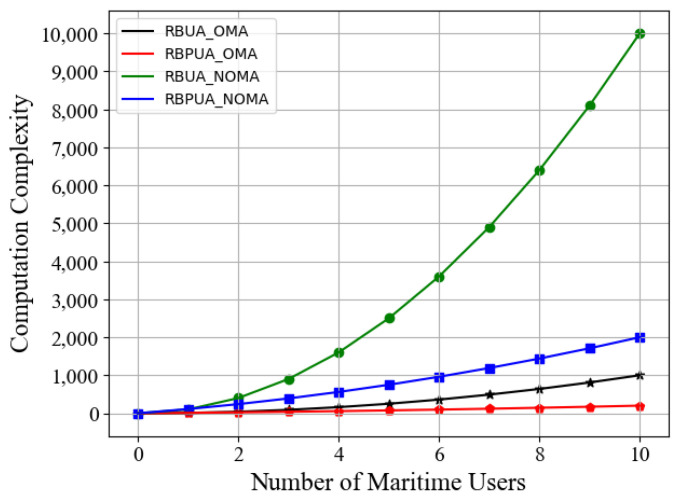
Computation complexity of RBPUA and RBUA schemes vs. number of users in NOMA or OMA communications.

**Figure 6 entropy-23-01454-f006:**
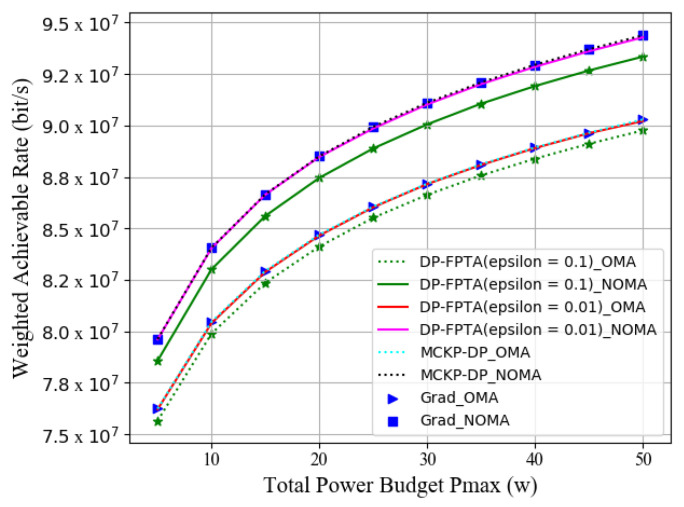
Weighted achievable rate of the three multi-subchannel joint optimization schemes vs. total power budget (line chart).

**Figure 7 entropy-23-01454-f007:**
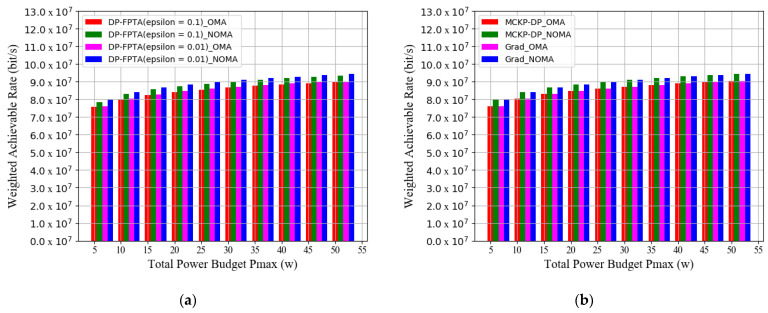
Weighted achievable rate of the three multi-subchannel joint optimization schemes vs. total power budget (histogram): (**a**) WAR of DP-PFTA with different *ε* and (**b**) WAR of MCKP-DP and GRAD algorithms.

**Figure 8 entropy-23-01454-f008:**
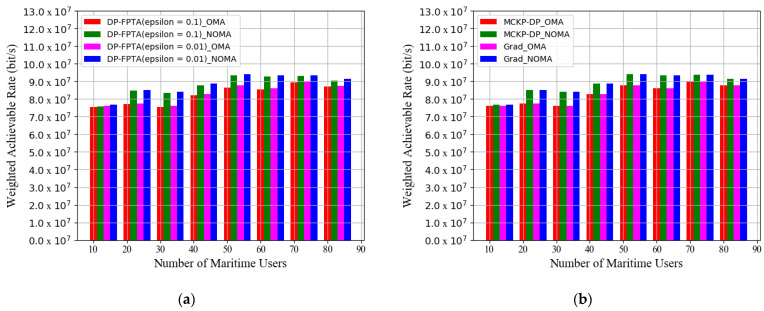
Weighted achievable rate of the three multi-subchannel joint optimization schemes vs. number of maritime users: (**a**) WAR of DP-PFTA with different *ε* and (**b**) WAR of MCKP-DP and GRAD algorithms.

**Figure 9 entropy-23-01454-f009:**
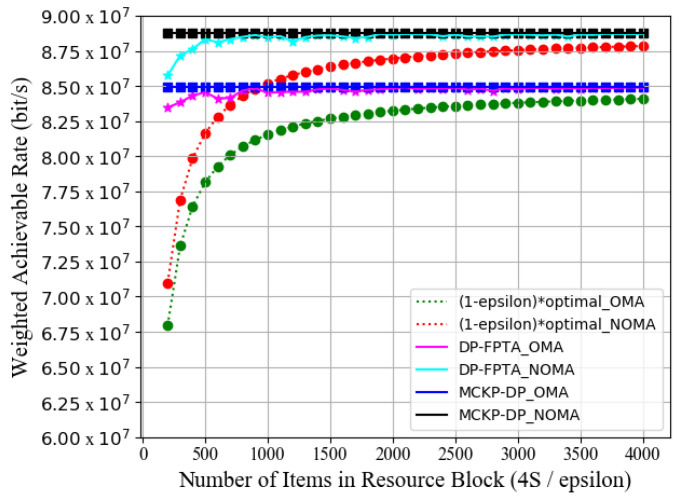
Weighted achievable rate of DP-FPTA and MCKP-DP vs. number of items in each resource block 4*S*/*ε*.

**Figure 10 entropy-23-01454-f010:**
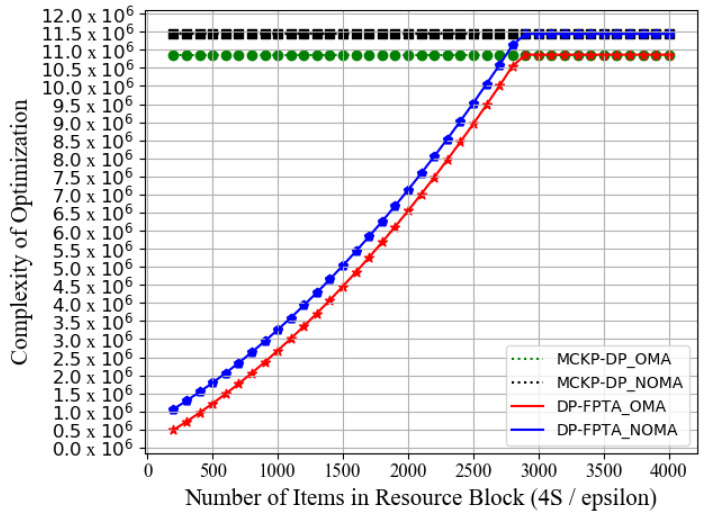
Computational complexity of DP-FPTA and MCKP-DP vs. number of items in each resource block 4*S*/*ε*.

**Table 1 entropy-23-01454-t001:** Reference value for the terrain irregularity parameter.

Irregular Terrain	Δ*h* (m)
Flat (or smooth water)	0
Plains	30
Hills	90
Mountains	200
Rugged mountains	500

**Table 2 entropy-23-01454-t002:** Suggested values for the electrical ground constants.

Medium	Relative Permittivity (F/m)	Conductivity (S/m)
Average ground	15	0.005
Poor ground	4	0.001
Good ground	25	0.020
Fresh water	81	0.010
Sea water	81	5.0

For most purposes, use the constants for an average ground.

**Table 3 entropy-23-01454-t003:** Suggested values of surface refractivity for different climates.

Climate	Surface Refractivity (N-Units)
Equatorial	360
Continental subtropical	320
Maritime subtropical	370
Desert	280
Continental temperate	301
Maritime temperate (over land)	320
Maritime temperate (over sea)	350

**Table 4 entropy-23-01454-t004:** Simulation parameters.

Symbol	Quantity	Value (Unit)
*d*	Radius of offshore area	5000 (m)
*f*	Carrier frequency	2.6 (GHz)
*σ*	Noise power spectral density	−174 (dB/Hz)
*B*	System channel bandwidth	5 (MHz)
*H_T_*	Height of BS transmitting antenna	15 (m)
*H_R_*	Height of maritime receiving antenna	5 (m)
Δ*h*	Terrain irregularity parameter	0 (smooth water)
TSite Criteria	BS deployment parameter	2 (very careful)
RSite Criteria	Receiver deployment parameter	0 (random)
radio_climate	Climate parameter	3 (tropical ocean climate)
*pol*	Antenna polarization method	1 (vertical polarization)
eps_permittivity	Relative permittivity of sea water	81 (F/m)
sgm_conductivity	Sea water conductivity	5 (S/m)
eno_refractivity	Surface refractivity	370 (*N*-units)^3^
*S*	Number of resource blocks	10
*T*	Number of maritime users	[10, 80]
*ξ*	Error tolerance (Grad)	10^−4^
*P* _max_	Total power budget	[5, 50] (W)
*J*	Number of power values (MCKP-DP)	10^3^
*p*	Minimum transmit power (MCKP-DP)	0.01 (W)
*ε*	Approximate error value (DP-FPTA)	0.1, 0.01
*A*	Maximum number of users multiplexed on a resource block	1 (OMA), 10 (NOMA)

## Data Availability

Not applicable.

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
