# Peer review of "Optimization Algorithms for Joint Power and Sub-Channel Allocation for NOMA-Based Maritime Communications"

_entropy, 2021, doi:10.3390/e23111454_

Round 1
Reviewer 1 Report
General comments:
The presented paper was devoted to a very important topic related to the reduction of latency in the RAN domain of 5G/6G systems in order to enable the provision of real-time services. In this case, it concerns maritime transport. Such services are also increasingly required in maritime communications. Using satellite connections does not guarantee low latency, if only because of long propagation distances. The proposed innovation may be applicable to massively connecting the EUs to the network of coastal base stations, due to the possibility of reducing the time needed to perform the calculations, therefore it is a good contribution to further work on a solution that will significantly increase the use of the future NOMA technique. As mentioned in the conclusions, a lot of research still needs to be done to adapt the proposed solutions in modern cellular systems. It is primarily about the MIMO technique and the effective work of the RAN domain in the C-RAN architecture. Another issue is the influence of the radio band used for communication and the influence of propagation phenomena (propagation model) occurring in other regions of the world, so that the proposal could be applied globally. The authors of the paper presented the results of analyzes and simulations for omnidirectional or sectoral communication. This is understandable due to the significant increase in the complexity of the analyzes, especially in the case of the SDM-based CoMP technique. However, such calculations should be performed in subsequent researches to demonstrate the limit of applicability of the method under specific variable channel conditions.
Detailed comments:
The paper was written in a concise and legible manner. It is worth describing the selected Longley-Rice propagation model in more detail in terms of the indicators used in it and discussing its applicability in individual radio bands that can be used for maritime communication, outside the selected 2.6GHz band. Also minor elements that should be improved:
- Line 42 – no italic font and index skew in the variable.
- In the initial formulas (1) and (2) the variables used are not described. These dependencies are commonly known, but a description (including any units) should appear (f, d, Ae, k1, k2, Ad, md, As, ms).
- Please organize the extensions of abbreviations, including the obvious ones, e.g. SIC.
- Figure 5 – in the scale description, please remove the so-called typo.
- In Appendix A the font used should be changed.
Summary:
The presented paper, with minor corrections and extensions, will be a good contribution to the knowledge of the properties and use of NOMA techniques in the future wireless communication interfaces.
Reviewer 2 Report
1. Expressions (6), (7), (10), (15), (17), (18), (19), (20-24), (28): you should number each row of the expressions. It could be confusing if you use one of rows under the same expression number. E.g. (28) instead F1...F4 use numbering of each expression (raw). Instead current expression (28) you should have 5 numbers.
2. Algorithm 2:
line 5 - Calculate the optimum...
It would be easier to follow the algorithm if you state the number of expression for optimization.
3. Algorithm 5:
Line 3 - how is the parameter (Greek letter) chosen? Computer probably uses a number which is defined by some function or heuristic by the programmer?
4. Algorithm 7:
Fully polynomial time approximation is used. It would be interesting to compare with other approximations and to conclude which one is the most suitable.
5. Figure 2:
If only 10m antenna has increased loss, why so much heights? If there are differences between 20-50, the graph should be zoomed to see differences.
6. You could comment:
https://doi.org/10.1155/2019/7842987
http://dx.doi.org/10.3390/app9204282
http://dx.doi.org/10.1109/ACCESS.2019.2926429
https://doi.org/10.1016/j.phycom.2021.101296
http://dx.doi.org/10.3390/s19235307
https://doi.org/10.3390/s21051833
Reviewer 3 Report
This paper entitled “Optimization Algorithms of Joint Power and Sub-channel Allo-2 cation for NOMA-based Maritime Communications” presents very interesting results concerning the WAR optimization for the NOMA-based offshore downlink communication scenario of the South China Sea, which can also be applied anywhere in the world in similar circumstances.
The paper is well written in English, well introduced, structured and referenced, including two appendixes. I can only see minor optimizations that can eventually me made, such as:
- The acronym for NP-hard in line 71 of page 2 was not previously introduced.
- In Equation (2), page 3, the value indicated for the free space loss is only true if the units of frequency (f) are in MHz and distance (d) are in km, however the units in this and other equations are not indicated.
In general, I consider the paper with very good quality, with impressive efficiency results, so I recommend it for publication.
Round 2
Reviewer 1 Report
General comments:
All the recommended corrections indicated in the previous version of the review were taken into account. Thank you for introducing corrections and explanations to the paper in accordance with the reviewer's recommendations. I have no further comments.
Summary:
The proposed algorithms and presented description of the simulation results will be a good contribution to the knowledge in area of wireless communications.
Reviewer 2 Report
Thanks for answers.
Reviewer 3 Report
I accept the paper in its present form to publication.
This manuscript is a resubmission of an earlier submission. The following is a list of the peer review reports and author responses from that submission.